# Modelling of Phase Structure and Surface Morphology Evolution during Compound Thin Film Deposition

**Gediminas Kairaitis and Arvaidas Galdikas *** 

Physics Department, Kaunas University of Technology, 50 Studentų st., LT-51368 Kaunas, Lithuania; gediminas.kairaitis@gmail.com
*** Correspondence: arvaidas.galdikas@ktu.lt

**Abstract:** The dependences of the surface roughness and the phase structure of compound thin films on substrate temperature and flux of incoming particles are investigated by a proposed mathematical model. The model, which describes physically deposited thin compound film growth process is based on the Cahn–Hilliard equation and includes processes of phase separation, adsorption, and diffusion. In order to analyze large temperature range and assuming deposition of energetic particles, the diffusion is discriminated into thermal diffusion, radiation-enhanced diffusion, and ion beam mixing. The model is adapted to analyze surface roughness evolution during film growth. The influences of the substrate temperature and incoming flux particles on the surface roughness are determined by a series of numerical experiments. The modelling results showed that the surface roughness increased as the substrate temperature rose. Besides, a similar relationship was discovered between substrate temperature and size of nanoparticles formed in binary films, so the increase in the surface roughness with the substrate temperature was attributed to the increase in size of nanoparticles.

**Keywords:** phase separation; kinetic modeling; thin films; surface roughness; compounds

## 1. Introduction

Nanocomposite materials, due to their promising and exceptional mechanical, thermal, optical, electrical, electrochemical, and catalytic properties, have attracted the great attention during the past few decades [1–6]. Those properties of nanocomposites may be notably different than those of the individual constituents [1]. Nanocomposites can exist in various morphological forms such as randomly or evenly distributed nanoparticles (of various shapes), columns, or layers [7]. Their multi-phase morphology may result in characteristics that are independent from the properties of each individual constituent present in the system, which provides a large range of applications [7] and creates the possibilities of varying their chemical and physical properties as a function of particle size, shape, and composition [2,8–11]. Since the phase structure determines the properties of nanocomposite materials, it is important to understand the influence of growth conditions (substrate temperature, growth rate, contents of depositing species, etc.) on the phase structure of films, which can provide a better understanding of how to control the phase structure of nanocomposites and reveal the mechanisms resulting in the formation of various phase structures. Besides the phase structure of a film, the surface roughness is another important characteristic affecting many properties of deposited films. The surface roughness influences many mechanical, optical, and electrochemical properties of thin films [1], and surface roughness is one of the characteristics that characterizes deposited thin films. The experimental measurement of surface roughness became common procedure for

characterizing deposited thin films properties. The surface roughness is highly influenced by the processes occurring during a thin film growth [12], so the relationship of this characteristic with the growth conditions can also provide more understanding about the growth mechanisms. Surface roughness of deposited thin films depend on various parameters such as deposition rate, temperature, surface pretreatment, composition, phase formation, etc. The control of surface roughness formation kinetics and understanding of the mechanisms influencing surface roughness is important in order to control deposited thin film properties and characteristics.

There have been several recent efforts to simulate the growth of nanostructured thin films based on kinetic Monte Carlo [13–15], molecular dynamics [16], and phase-field [17–20] approaches. The influences of substrate temperature [13], deposition rate [13,15], substrate tilt angle [14], and composition ratios [15] on the structure of nanocomposites have been recently determined by using the kinetic Monte Carlo approaches. It was discovered in [14] by Bouaouina et al. that an increase in the substrate tilt angle resulted in an increase in the surface roughness and the tilt angle of TiN columns. In [15], it was shown that, at relatively low growth rates and thinner deposited multilayers, the grown thin films had the structure of a columned form, whereas higher growth rates resulted in the formation of the dotted structures. The factors responsible for the stress generation during a growth of body-centered cubic (BCC) metal thin films were investigated by using a molecular dynamics approach [16] by Zhou et al. In [16], the influences of the surface morphology, the coalescence of adjacent islands, the injection energy, the grain size, and the film texture on the stresses generated in the thin films were determined and discussed. The growth conditions and mechanism resulting in the formation of self-organized alternating layers in metal: carbon thin films were determined and explained in [17]. The effects of deposition rate and substrate temperature during the growth of Cu–Mo films were investigated by Derby et al. [18] and Ankit et al. [19]. The morphology map containing three distinct phase structures (vertical composition modulations, lateral composition modulations, and random composition modulations) with respect to deposition rate and mobility has been provided by Ankit et al. [19]. The effects of deposition rate, dissimilar bulk and surface kinetics, phase fraction, and dissimilar elastic response on the resulting microstructure were determined by Stewart and Dingreville [20].

This work proposes a model for growth of two-phase thin film that can model the evolution of the surface roughness. The proposed model uses the Cahn–Hilliard equation to model the phase separation during a thin film growth. The main objective of this work is to investigate the influence of substrate temperature, flux of incoming to the surface particles on the phase structure, and surface roughness of biphasic thin films. The dependences of surface roughness on substrate temperature and flux of incoming particles are determined.

## 2. Methods

A three-dimensional grid is used to simulate the distribution of thin film components and substrate material during a film growth. Each grid point is associated with three local relative concentrations $c_A^{i,j,k}$, $c_B^{i,j,k}$, $c_S^{i,j,k}$, $i = 1, \ldots, I, j = 1, \ldots, J, k = 1, \ldots, K$, which denote the concentrations of component $A$, $B$, and substrate material, respectively, at grid point $i, j, k$. The indices $i, j$ denote the positions of a respective grid cell in the horizontal directions, $k$ denotes the position of a respective grid cell in the vertical direction (the film growth direction). For each grid cell the condition $c_A^{i,j,k} + c_B^{i,j,k} + c_S^{i,j,k} \leq 1$ is always satisfied. Since our main objective is to model the distribution of thin film components and the surface roughness during a film growth, the surface diffusion is a very important process to address. The phase separation of both components occurring through the surface diffusion is described by using the Cahn-Hilliard equation. Therefore, the changes in component $A$ concentration due to phase separation in the surface layer of a growing film are described through the following equation:

$$\frac{\partial c_{A_S}}{\partial t} = \nabla D_A \nabla \left( \frac{\mathrm{d}f(c_{A_S})}{\mathrm{d}c_{A_S}} - \gamma \nabla^2 c_{A_S} \right) \tag{1}$$

where $c_{A\_S}$ ($c_{A\_S}^{i,j}$, $i = 1, \dots, I$, $j = 1, \dots, J$) denotes the concentration of component $A$ in the surface layer, $D_A$ is the diffusion coefficient, $f$ is the free energy density of a homogenous system (function $f$ used in this work is same as in [21]), and $\gamma$ is the power coefficient of phase gradient.

The concentrations of component $A$ in the surface layer $c_{A\_S}^{i,j}$ are defined through the following equation:

$$c_{A\_S}^{i,j} = c_A^{i,j,k^*} + c_A^{i,j,k^*-1}(1 - c_A^{i,j,k^*} - c_B^{i,j,k^*} - c_S^{i,j,k^*}), \tag{2}$$

where $k^*$ is the highest position (in the growth direction) at which $c_A^{i,j,k} + c_B^{i,j,k} + c_S^{i,j,k} > 0$ is satisfied. The term $c_A^{i,j,k^*} + c_B^{i,j,k^*} + c_S^{i,j,k^*}$ may take any value between 0 and 1, so the use of Equation (2) for defining concentrations in the surface layer is necessary. The changes in component $B$ concentration in the surface layer are also described by Equation (1), but in order to do this, the term $c_{A\_S}$ in Equation (1) should be replaced with $c_{B\_S}$ (the concentration of component $B$ in the surface layer), which is defined analogously to Equation (2). So, two Cahn–Hilliard equations are solved to determine the changes in the concentrations of both components due to phase separation in the surface layer of a growing film.

Equation (1) is also used to describe another process. This process ensures that atoms of type $A$ adsorbed on a nano-island made of same type atoms gather together to form the surface pattern of that nano-island. This process is described by the same equation as Equation (1) but with the following variable $c_{AS}^{i,j}$ (instead of that given by Equation (2)) and the diffusion coefficient $D_{1A}$ (instead of $D_A$) in it:

$$c_{AS}^{i,j} = c_A^{i,j,k^*}, \tag{3}$$

where the meaning of $k^*$ in Equation (3) is the same as in Equation (2). The values of $c_A^{i,j,k^*}$ $i = 1, \dots,$ $I, j = 1, \dots, J$ represent the concentrations of component $A$ in the all surface cells, and each of them varies in the interval from 0 to 1. The process described by Equation (3) ensures that particles from cells with relatively low values of $c_A^{i,j,k^*}$ can gather together to form fully filled surface cells. $D_{1A}$ is the diffusion coefficient. The value $D_{1A} = D_A$ is used on nano-islands made of component $A$, $D_{1A} = 0$ is used anywhere else. The same process is defined for component $B$, its definition is analogous to Equation (3).

The next process included in the model is the diffusion of atoms on the nano-island due to its surface curvature. The changes in component $A$ concentration due to the surface curvature of a growing film are described through the following equation:

$$\frac{\partial c_{A\_S}}{\partial t} = \nabla D_A \nabla\left(-p_A c_{A\_S} \nabla^2 h\right), \tag{4}$$

where the term $\nabla^2 h$ represents the surface curvature (this expression has been chosen for simplicity), $h$ denotes the position of the thin film surface, and $p_A$ is the proportionality coefficient. Equation (4) ensures that atoms of respective type diffuse from crests to valleys of nano-islands due to its surface curvature. This process always results in smoothing of thin film surfaces. Equation (4) is also used to determine the changes in component $B$ concentration due to the surface curvature, but $c_{B\_S}$ and $p_B$ should be used instead of $c_{A\_S}$, $p_A$, respectively, in Equation (4).

Changes in concentrations of both film components (at highest occupied surface cells $c_A^{i,j,k^*}$) due to the adsorption process are described by using the following equations:

$$\frac{\partial c_A^{i,j,k^*}}{\partial t} = k_{AA} i_A c_{AS}^{i,j} + k_{AB} i_A c_{BS}^{i,j} + k_{AS} i_A c_{SS}^{i,j},$$

$$\frac{\partial c_B^{i,j,k^*}}{\partial t} = k_{BA} i_B c_{A\_S}^{i,j} + k_{BB} i_B c_{B\_S}^{i,j} + k_{BS} i_B c_{S\_S}^{i,j}, \tag{5}$$

$i = 1, \ldots, I, j = 1, \ldots, J, k_{AA}, k_{AB}, k_{AS},$ are the sticking coefficients ($k_{AB}$ denotes the sticking coefficient of $A$ type atoms to the surface made of component $B$), $i_A, i_B$ are relative fluxes of both film components, and $c_{A\_S}^{i,j}, c_{B\_S}^{i,j}, c_{S\_S}^{i,j}$ are the surface concentrations of component $A, B,$ and substrate material, respectively. The sum of all six products $(k_{AA}i_A c_{A_S}^{i,j} + k_{AB}i_A c_{B_S}^{i,j} + k_{AS}i_A c_{S_S}^{i,j} + k_{BA}i_B c_{A\_S}^{i,j} + k_{BB}i_B c_{B\_S}^{i,j} + k_{BS}i_B c_{S\_S}^{i,j})$ amounts to the total growth rate of the thin film.

All four equations describing the considered processes with respect to growing phase are added together to get the final model, and this model is used to simulate the growth and the kinetics of the phase structure. Since we investigate two growing phases, two models consisting of Equations (1)–(5) with respect to each growing phase are considered. It is assumed that the flux of incoming particles or the phase separation of deposited species do not cause any changes in the substrate. To simulate the evolution of the system during a thin film growth, the equations of the model are solved in time by using the explicit finite difference method.

Our main objective is to investigate the dependence of the roughness and the phase structure of thin films on substrate temperature. In order to analyze large temperature range, the diffusion is discriminated into thermal diffusion, radiation-enhanced diffusion, and ion beam mixing. In the case of the physical vapor deposition (PVD) process when energetic particles arrive to the surface, the mixing process can occur, which influences the diffusion [22–25]. Generally, three regimes are discriminated [26]: (1) Below $0.2T_m$ ($T_m$ is melting temperature of given material), the influence of thermal diffusion is negligible and the mixing process dominates. The diffusion in that regime does not depends on temperature and depends only on flux and energy of arriving particles, the dependence on flux is linear. (2) Between $0.2T_m$ and $0.6T_m$, the influence of thermal diffusion becomes significant, and diffusion becomes dependent on temperature but still remains dependent on flux and energy of arriving particles. Diffusion in that regime depends on both thermal and radiation defects. The concentration of radiation defects depends on ion flux density as square root function. (3) Above $0.6T_m$, the thermal diffusion significantly exceeds the mixing and the diffusion becomes dependent only on temperature. The influence of energy and flux of arriving particles becomes negligible. Assuming this the dependence of the diffusion coefficient $D_A$ on the substrate temperature, $T$ is expressed as follows:

$$D_A = \begin{cases} D_m = \alpha i, \ T < 0.2T_m, \\ D_{Rad} = \sqrt{i}\beta e^{-\frac{Q_1}{RT}}, \ 0.2T_m < T < 0.6T_m \\ D_{Th} = \delta e^{-\frac{Q_2}{RT}}, \ T > 0.6T_m \end{cases} \tag{6}$$

where $i = i_A + i_B$ is the total flux of components $A$ and $B$, $\alpha, \beta,$ and $\delta$ are the proportionality coefficients (values of 0.025, 0.225, 2 are used in our calculations, respectively), $Q_1, Q_2$ are the activation energies of radiation enhanced diffusion and thermal diffusion, respectively (10 kJ/mol and 40 kJ/mol are used, respectively). In calculations, the melting temperature $T_m = {\sim}3000$ K (value close to the melting temperature ceramics) is used. The plot of diffusion coefficient vs. substrate temperature used in calculations is given in Figure 1.

Assuming the Langmurian temperature dependence of the adsorption process, the following function of the growth rate is used in our modelling:

$$V = \Sigma k_{jl} i c_{jl}, \ k_{jl} = \frac{c_1}{\sqrt{T}} - c_2 e^{-\frac{Q_3}{RT}}, \tag{7}$$

where the first term in the second relationship is related to the process of adsorption, and the second term is related to the process of desorption, $i$ is the total relative flux, $c_1, c_2$ are the proportionality coefficients (values of 60 and 2.5 are used, respectively), and $Q_3$ is the activation energy of desorption. $Q_3 = 10$ kJ/mol is used.

Figure 2 shows the plot of growth rate vs. substrate temperature (obtained by using relative flux of $i = 0.8$ s$^{-1}$) used in calculations.

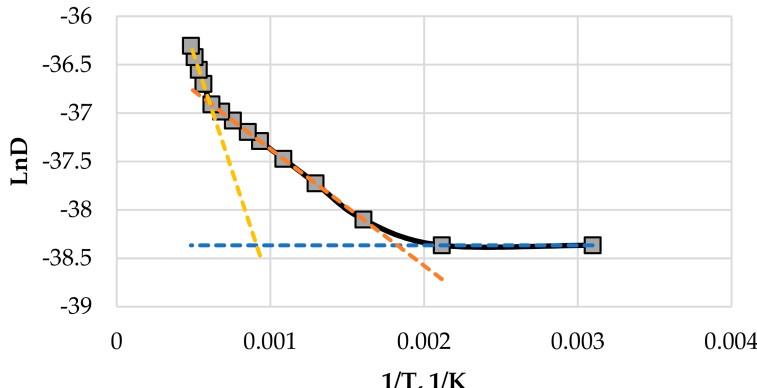

**Figure 1.** The dependence of diffusion coefficient on substrate temperature used in calculations.

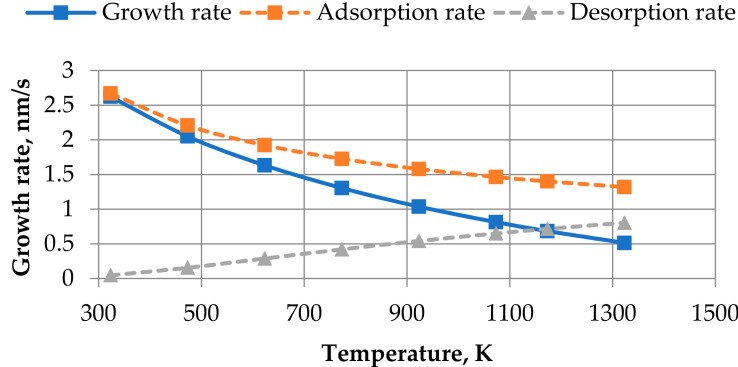

**Figure 2.** The dependence of total growth rate (solid line) on substrate temperature used in calculations. By dot lines, the partial components of growth rate (Equation (7)), adsorption, and desorption rates are shown.

## 3. Results and Discussion

All calculations are started from a perfectly flat substrate (the substrate size is 36 nm × 36 nm). Initially, the substrate occupies five layers in the computational grid and on that substrate, there is a thin layer made of a mix of components *A* and *B* where concentrations of both components are uniformly distributed in the interval {0, 0.01}. This randomly chosen layer of a mix of components *A* and *B* is called an initial condition. Since there are no random terms in the equations describing growth process (given in Equation (5)), the initial condition with a random distribution of thin film components is necessary. Calculations will be performed in the temperature range from 323 to 923 K. The temperatures at which the growth of thin films will be modelled are marked with square points in Figures 1 and 2 (five rightmost points in Figure 1 and five leftmost points in Figure 2). The values of parameters $p_A = p_B = 1.6 \times 10^{-13}$ J/m (see Equation (4)), $D_B = D_A$ (see Equation (5) and Figure 1), and $\gamma = 4.8 \times 10^{-13}$ J/m (see Equation (1)) are used in calculations. All six sticking coefficients (see Equation (5)) are assumed equal between themselves, and their dependencies on the substrate temperature are identical to that of the adsorptions rate given in Figure 2.

Figure 3 shows the plots of surface roughness $R_q$ vs. substrate temperature calculated by using different relative fluxes of particles arriving at the surface. The relative concentrations of components in simulated thin films are $c_A = 30\%$, $c_B = 70\%$. Any point in Figure 3 is an average roughness value from three calculations using different initial conditions. From Figure 3, it is seen that the surface roughness $R_q$ increases with the increase in the substrate temperature. This tendency is observed for all relative fluxes that were examined. From Equations (6) and (7), it is seen that changes in the relative

flux affect both values of diffusion coefficients and growth rate (the relative flux is directly proportional to both of those values).

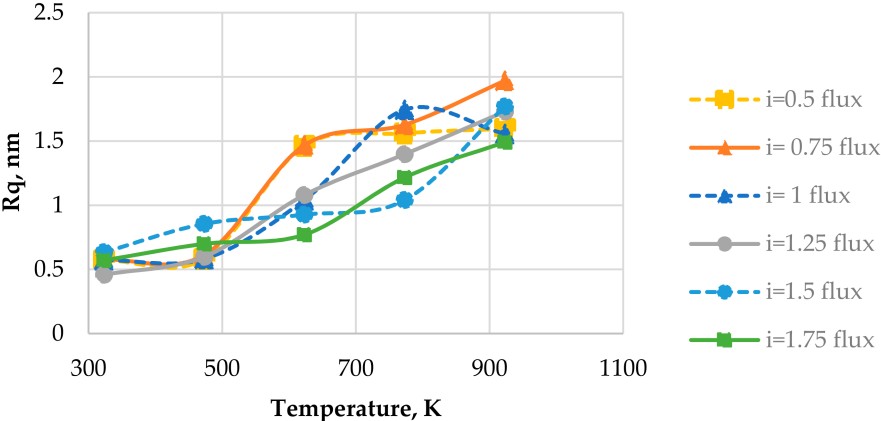

**Figure 3.** Plots of surface roughness vs. substrate temperature with different relative fluxes of arriving particles.

Figure 4 shows the plot of nano-particle size of phase *A* in the horizontal directions (or directions parallel to the substrate surface) vs. substrate temperature at relative flux of $i = 1$ s$^{-1}$. From Figure 4, it is seen that the average nano-particle size increases linearly from 7 to 12.5 nm as the substrate temperature rises from 323 to 923 K. From Figures 3 and 4, one can notice that both the surface roughness $R_q$ and the average nano-particle size increase monotonously as the substrate temperature rises. The similar relationships between substrate temperature and nano-particle size were also reported in other experimental studies [27,28] and theoretical works [29,30].

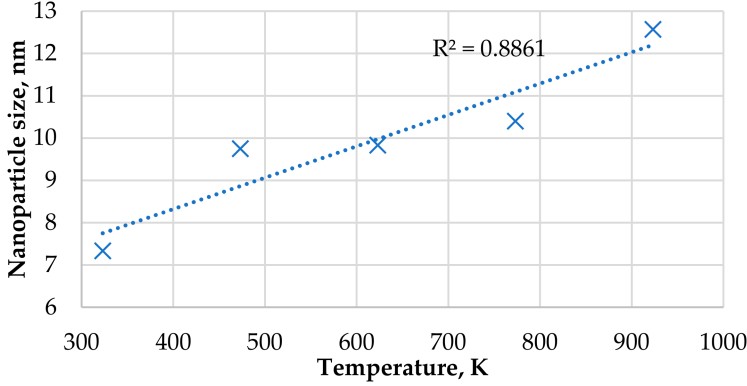

**Figure 4.** Plot of nano-particle size of phase *A* (in the horizontal directions) vs. substrate temperature at the relative flux of arriving particles of $i = 1$ s$^{-1}$.

Figure 5 shows pairs of the surface maps (a, b), the concentrations plots of component *A* in the surface layer (c, d), and cross-sectional views (e, f) obtained with different substrate temperatures of 323 K (a, c, e) and 923 K (b, c, f) and constant relative flux of $i = 1$ s$^{-1}$. The cross-sectional views are given through the plane $y = 18$, which is marked in the concentration plots. Brown color in Figure 5a,b indicates regions of the thin films in which their thicknesses are greatest. Since the substrate initially occupies five layers (layer thickness is 1 nm) in the computational grid and 5 nm is the lowest thickness observed in images given in Figure 5a,b, blue color in those images marks regions of the substrate, which are still not covered with either phase *A* or phase *B*. Brown color in Figure 5c,d indicates regions of the surface of the thin films, which are made of phase *A* (the relative concentration of *A* is 1.0).

Blue color in Figure 5c,d marks regions that are made of either phase *B* or the substrate material, i.e., regions free of component *A* (in those images substrate material can be distinguished from phase *B* by using the respective surface maps given in Figure 5a,b). Cyan color in Figure 5e,f represents regions made of phase *A*; yellow color in those images marks regions made of phase *B*. It should be noted that the average nano-particle size (given in Figure 4) was assessed from the concentration plots of component *A* that are partially given (for two substrate temperatures) in Figure 5c,d.

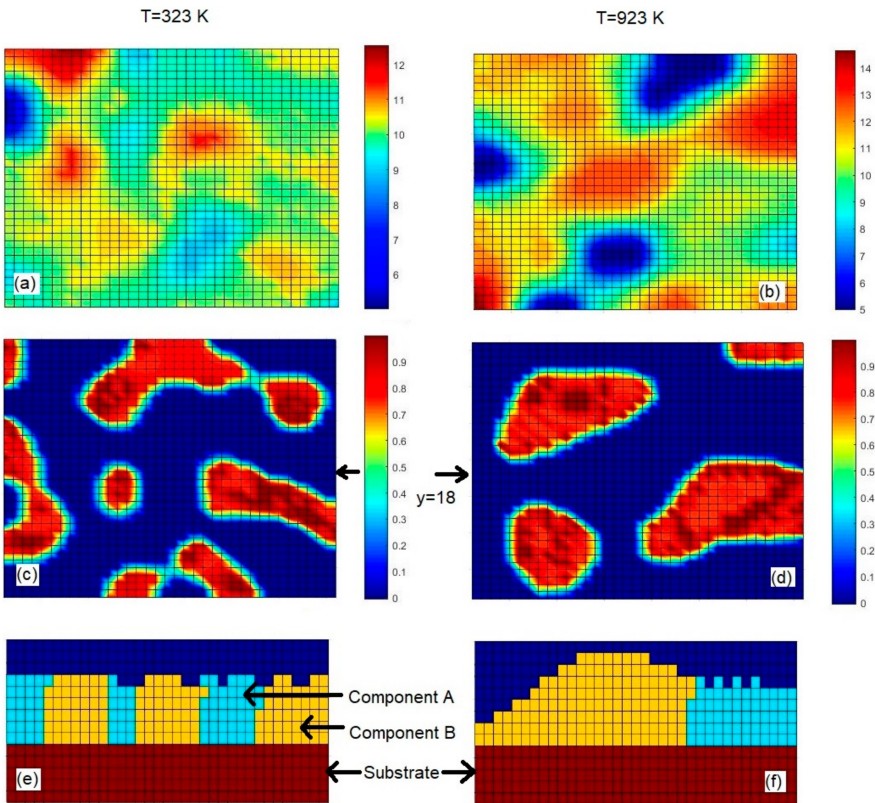

**Figure 5.** The surface maps (**a**,**b**), concentration plots of component *A* (in the surface layer of thin films) (**c**,**d**), and cross-sectional views (**e**,**f**) calculated at the substrate temperatures of 323 K (**a**,**c**,**e**) and 923 K (**b**,**c**,**f**).

From Figure 5c,d, it is clearly seen that the increase in the substrate temperature results in visually larger nanoparticles of phase *A* and a smaller amount of those nanoparticles. From cross-sectional views given in Figure 5e,f, it is seen that the lower substrate temperature causes the lower variation in the thin film thickness. This statement is also supported by the data given in Figure 5a,b. From Figure 5a,b, it is seen that at the substrate temperature of 323 K the thin film thickness varies in the range from 5 nm to ~12.5 nm (including the thickness of 5 nm of the substrate material), whereas at the substrate temperature of 923 K, the thin film thickness varies between 5 nm and ~14.5 nm. In addition, at the higher substrate temperature, there are more blue colored regions in Figure 5b (in comparison to Figure 5a), which indicate areas that are still not covered with either phase *A* or phase *B*. Those blue colored areas also contribute to the higher surface roughness observed at the higher substrate temperature. From Figure 5, one can deduce that, with given model parameters, the larger nano-particle size (caused by the higher substrate temperature in this case) results in a larger variation in the thickness of the film, which causes a higher surface roughness. The increase in the surface roughness $R_q$ with increasing substrate temperature observed in Figure 3 can be attributed to the increase in the average nano-particle size (with increasing substrate temperature), because in all calculations presented in Figure 3 both the average grain size and surface roughness increase as the substrate temperature rises. Figure 4 (revealing the linear relationship between grain size and substrate temperature) shows only one case of relative flux of $i = 1$ s$^{-1}$. The influence of substrate temperature on both crystallite size and

surface roughness is also observed in samarium doped ceria thin films grown by e-beam physical vapor deposition [31], where both the crystallite size and the surface roughness of the thin films increase with temperature of the substrates.

Figure 6 shows the evolution in time of the surface roughness of the thin film grown at the substrate temperature of 773 K with the relative particle flux of $i = 1$ s$^{-1}$. The curve passes maximum; at the beginning of deposition process, surface roughness increases and then starts to decrease. The similar kinetics of surface roughness are obtained by using Mote Carlo simulation techniques [32]. From Figure 6, it is seen that the surface roughness most rapidly increases during first two seconds of the film growth. From 2 s to 3.2 s, the surface roughness increases slower with time in comparison to first two seconds. At the growth time of ~3.25 s, the surface roughness reaches its maximum value and stars to decrease thereafter. The surface roughness decrease rate is highest during the period between 3.5 s and 7 s; after this period, the surface roughness starts to decrease slower and only negligible surface roughness oscillations are observed during the last few seconds.

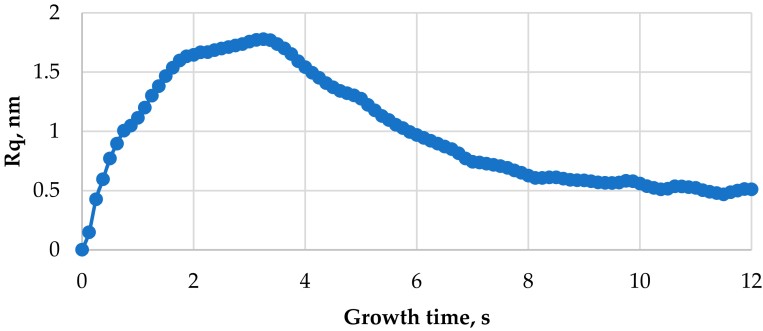

**Figure 6.** Plot of surface roughness vs. growth time at the substrate temperature of 773 K and the relative flux $i = 1$ s$^{-1}$.

In order to understand the kinetics of surface roughness shown in Figure 6, the surface maps at different typical times $t = 0.625$ s (before maximum), $t = 3.25$ s (maximum point), $t = 6.375$ s (after maximum), and $t = 12$ s (last point in Figure 6) are given in Figure 7. At times $t = 0.625$ s and $t = 6.375$ s, the surface roughness $R_q = 0.895$ nm is the same. Colormaps are given next to the respective surface map. Brown color in all surface maps given in Figure 7 indicates regions of the thin films in which their thicknesses are greatest. Blue color marks regions in which the thicknesses of films are lowest; in Figure 7a,b, substrate regions that are not covered with either phase $A$ or phase $B$ are marked with blue color. Phases of components $A$ or $B$ cannot be distinguished from maps given in Figure 7. Concentration plots of both components are needed in order to distinguish those two phases. From Figure 6, it can be seen the surface maps given in Figure 7a,c correspond to two cases at which the surface roughness of ~0.895 nm is the same. The surface map given in Figure 7b corresponds to the moment at which the highest value of the surface roughness is reached. From Figure 7a, it is seen that at an early growth time, there is a group of nano-islands formed on the substrate. Less than half of the substrate area is covered with nano-islands. The diameters of nano-islands vary in the range from 5 to 8 nm. The height of the highest nano-island is about 3.5 nm. In Figure 7b, we observe larger and higher nano-islands, some of them are already connected to each other. The height of the highest nano-island is about 5.5 nm at the growth time of 3.25 s. From Figure 7b, it is also seen that there are substrate areas that are still not covered with either phase $A$ or $B$ (marked with blue color in Figure 7b), and blue and brown areas in Figure 7b occupy pretty similar fractions of the surface area, which results in the relatively high surface roughness observed at $t = 3.25$ s. After $t = 3.25$ s, phases made of components $A$ or $B$ grow relatively fast on those uncovered substrate areas (blue regions in Figure 7b) in comparison to the tops of nano-islands. From the surface map, taken ~3.1 s later (at $t = 6.375$ s) that is given in Figure 7c, it is seen that the whole substrate is covered with phases made of depositing species. In Figure 7c, there are no such deep valleys between growing nano-islands, as it is seen in Figure 7b.

This also explains the lower value of the surface roughness observed at $t$ = 6.375 s. From Figure 6, it is seen that the surface roughness $R_q$ decreases from $t$ = 3.25 to ~10 s and starts slightly oscillating thereafter. The final surface map is given in Figure 7d. In this surface map, we observe a minimum variation in the thickness of the film in comparison to all other surface maps given in Figure 7.

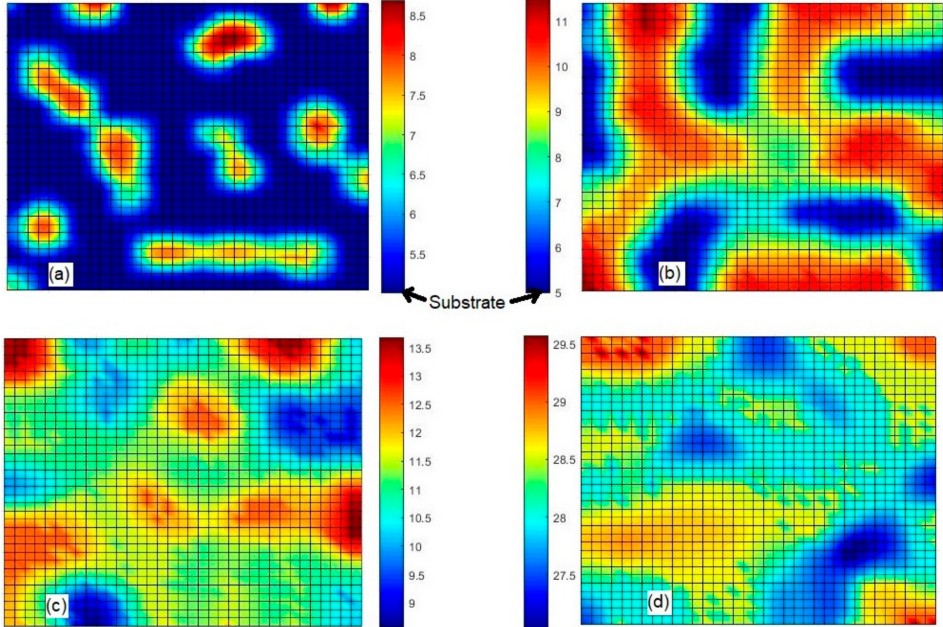

**Figure 7.** Surface maps of the thin film grown at the temperature 773 K with the relative flux $i$ = 1 s$^{-1}$ taken at $t$ = 0.625 s (**a**), $t$ = 3.25 s (**b**), $t$ = 6.375 s (**c**), and $t$ = 25 s (**d**).

Figure 8 shows the plots of surface roughness as a function of relative flux at different substrate temperatures. Figure 8 uses the same data that were used in Figure 3. The average surface roughness for substrate temperatures of 623 K, 773 K, and 923 K ($>0.2T_m$) is also given in Figure 8. From Figure 8, it is seen that the flux density does not have any consistent influence on the surface roughness at the substrate temperatures of 323 K and 473 K ($<0.2T_m$). The presence of flux density $i$ in the dependencies of diffusion coefficient and growth rate on the substrate temperature is the reason for a weak influence of the flux density on the surface roughness at two lowest considered temperatures. From Equation (6), it is seen that the flux $i$ is directly proportional to the diffusion coefficients $D_m$. From Equation (7), it is seen that the flux density $i$ is directly proportional to the growth rate. It is known that an increase in the diffusion coefficient alone results in an increase in the average size of nanoparticles in the horizontal directions. Since the surface roughness correlates to the average size of nanoparticles in the considered case, an increase in the diffusion coefficient alone results in an increase in the surface roughness. However, it is also known that an increase in the growth rate alone results in a decrease in the average size of nanoparticles. The correlation between the average size of nanoparticles and the surface roughness is retained in this case, so an increase in the growth rate alone results in a decrease in the surface roughness. This is the opposite effect in comparison to that of an increase in the diffusion coefficient alone, so an increase in the flux density $i$, which causes the linear increases in the diffusion coefficient and the growth rate, induces two counteracting effects and has no significant influence on the surface roughness at the two lowest considered substrate temperatures.

From Figure 8, it is seen that the average surface roughness of the cases of three highest substrate temperatures monotonously decreases as the flux density rises from 0.75 to 1.75 s$^{-1}$. A small increase in the average surface roughness is observed when the flux density is changed from 0.5 to 0.75 s$^{-1}$. At the three highest substrate temperatures considered, the diffusion coefficient $D_A$ is defined as $D_{Rad}$ in Equation (6). In this case, only the square root of flux density $\sqrt{i}$ is directly proportional to the

diffusion coefficient. This effect causes a lower increase (in comparison to the cases of two lowest substrate temperatures) in the diffusion coefficient as the flux density rises. A greater value of the flux density $i$ induces a stronger effect. The flux density remains directly proportional to the growth rate at three highest substrate temperatures, so at these substrate temperatures, the effect of an increase in the growth rate starts to prevail over the effect caused by an increase in the diffusion coefficient, which results in the decrease in the average surface roughness as the flux density rises.

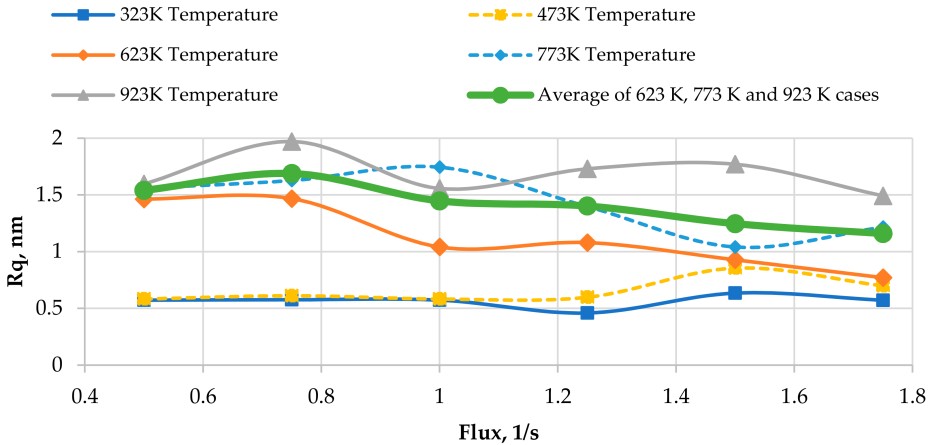

**Figure 8.** Plots of surface roughness vs. relative flux for different substrate temperatures.

Figure 9 shows three plots of the surface roughness vs. substrate temperature at the relative flux $i = 0.5 \text{ s}^{-1}$ obtained by using three different initial conditions. Points obtained by using the same initial condition are connected to each other with the same line type. The average of three given curves is presented in Figure 3 (see the curve obtained by using $i = 0.5 \text{ s}^{-1}$). From Figure 9, it is seen that an initial condition may significantly affect the dependence of surface roughness on temperature. Values of $R_q$ corresponding to the first and the third experiments mostly increase as the substrate temperature rises, but the curve corresponding to the second experiment shows different behavior. In this case, the slight decrease in the surface roughness is observed as the substrate temperature varies from 623 K to 923 K.

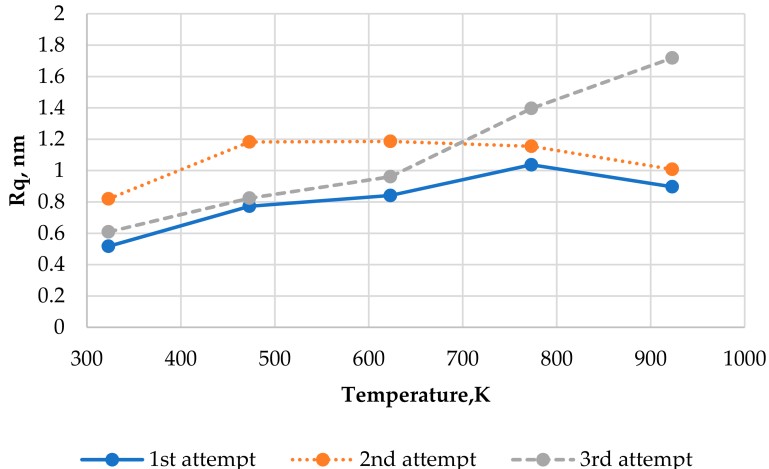

**Figure 9.** Three plots of the surface roughness vs. substrate temperature at the relative flux $i = 0.5 \text{ s}^{-1}$ obtained by using three different initial conditions.

## 4. Conclusions

In this work, the phase-field model for simulating the evolution of the phase structure and the surface morphology during compound thin film growth is proposed. The influences of substrate

temperature and flux of incoming particles on the surface roughness and the phase structure of thin films were explored by using the proposed mathematical model. The modelling results showed that the surface roughness of thin films increased with an increase in the substrate temperature. A similar relationship was noticed between the size of nanoparticles formed and the substrate temperature. Therefore, the increase in the surface roughness caused by an increase in the substrate temperature was attributed to the increase in the size of nanoparticles. No relationship between the flux of incoming particles and the surface roughness was reported by our modelling results for lower than $0.2T_m$ substrate temperatures. For higher than $0.2T_m$ substrate temperatures, a higher flux density resulted in a decrease in the surface roughness in most cases. This effect became stronger when a relatively higher value of the flux density was used.

**Author Contributions:** Conceptualization, G.K. and A.G.; methodology, G.K. and A.G.; software, G.K.; validation, G.K. and A.G.; investigation, G.K. and A.G.; writing—original draft preparation, G.K. and A.G.; writing—review and editing, G.K. and A.G.; visualization, G.K. and A.G.; supervision, A.G.; funding acquisition, G.K. and A.G. All authors have read and agreed to the published version of the manuscript.

**Funding:** This project has received funding from European Regional Development Fund (project No 01.2.2-LMT-K-718-01-0071) under grant agreement with the Research Council of Lithuania (LMTLT).

**Acknowledgments:** Authors would like to express their gratitude for the following individuals for their expertise and contribution to the manuscript: G. Laukaitis, T. Moskaliovienė, K. Bočkutė, M. Galdikas, D. Virbukas, M. Sriubas, and V. Kavaliūnas.

**Conflicts of Interest:** The authors declare no conflict of interest.

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
