# Peer review of "Modelling of Phase Structure and Surface Morphology Evolution during Compound Thin Film Deposition"

_coatings, doi:10.3390/coatings10111077_

Round 1

Reviewer 1 Report

The article is devoted to a very important and hard topic. In many cases of thin film growth an empirical search of optimal technology conditions is inevitable because an each pair film-substrate has the unique set of parameters: crystal lattices of film and substrate, thermal coefficients, substrate roughness, temperatures of adsobtion and reevaporation of components and so on. Moreover, deposition conditions - temperature, flux density, growth rate always affect on thin film properties. Based on above factors, the modeling of film growth is a very complex task.
On the reviewer opinion, in the presented text the following points should be clarified:

Authors are very thankful to reviewers for valuable suggestion to improve our work.

1. The field of applicability of the model proposed should be determined. Does it applicable to all binary thin films independent on possible chemical interaction between the components or between the components and the substrate?

  • Author Response:Specific systems may have some specific aspects, which can be incorporated into model when they are considered. In model the main processes taking place during binary films deposition are involved. The possible chemical interaction between components of the films and between film components and the substrate are involved through the free energy density function (eq.(1)) which defines solubility of film components and values of sticking coefficients. Model involves general aspects of binary films growth.

2. The substrate roughness is not included in the model. But this factor is always presented in a real situation. In this case, the model is only mathematical and very far from the reality.

  • Author Response:Surface roughness is included into the model indirectly. It is not as parameter but is the result of 3d films growth. Particles are deposited on rough surface. The main purpose of this work is surface roughness calculations.

3. The absence of the dependance of surface roughness on the flux density looks very strange concerning real films on real substrates. Moreover, the flux dencity directly affect on supersaturation and hence, on island form and dimencions, on growth rate, on crystallite sizes and on roughness.

  • Author Response:This conclusion arises because, for the first assumption, we assume that both, deposition rate and diffusion coefficients (mixing and RAD) linearly depends on flux density. To perform a better investigation of flux density influence on the surface roughness, we adopted another relationship between diffusion coefficient and flux density at substrate temperatures 0.2Tm < T < 0.6Tm. According to this relationship, square root of flux density is directly proportional to diffusion coefficient at the given interval of temperatures (see Eq. (6)). The modified relationship is adopted when there are radiation defects generated due to ion bombardment. Results given in Figures 3 and 8 were recalculated using the new relationship. New effects of flux density are discussed and explained in lines 308-335. Conclusions about the influence of flux density were also changed (see lines 360-364).

4. What is the reason of 30/70 component composition choosing? Will the situation be different for 50/50 or 70/30 relation?

  • Author Response:There are no special reasons for 30/70 component composition choosing. We analyze binary system with near equal composition. Because of function f in Eq. (1) 30/70 or 70/30 results would be the same. At different compositions quantitatively results will differ, but not qualitatively and obtained dependencies would remain the same.

5. The mismatch between fig. 6 and line 266 concerning the last point in time scale should be corrected.

  • Author Response:This value of last point in time scale has been corrected.

Reviewer 2 Report

Referee’s report on the manuscript “Modelling of Phase Structure and Surface Morphology Evolution During Compound Thin Film Deposition” by Gediminas Kairaitis and Arvaidas Galdikas.

This manuscript analyzes the growth of a compound thin film by means of finite element modelling of the deposition. As the main result, authors conclude that an increase of the growth temperature produces an increase of the particle size as well as a higher roughness on the film surface. The article is well written and clear in most aspects, although there are some relevant issues that, to my opinion, have to be addressed in the text before suggesting its acceptance for publication.

Authors are very thankful to reviewers for valuable suggestion to improve our work.

1. Most of my concerns relate to the model development and assumptions. For instance, the scale of the simulation is about 10 nm. How good is a finite element model to analyze such small scale lengths. Would not be the atomic discretization relevant in such small scales? What is the size of the cell element in the simulations? Moreover, it seems that solutions have much more resolution than the cells in figure 5 (top views). I think authors should discuss this issue in the manuscript.

  • Author Response:The model uses the finite difference method, not the finite element method. There are no numerical problems with the method used.

The atomic discretization could be relevant in such scales, but we use 1 nm size of the cell element in the simulations to reduce computational time.

It is an impression. We just use some interpolation to make transitions (from one color to another) smoother in those plots. Actually, solutions do not have more resolution than the density of cells in figure.

2. Authors consider particle fluxes and substrate temperature as two independent quantities. However, in practical conditions, an increase of substrate temperature would produce a rarefaction of the gases near the substrate region, thus affecting the fluxes. Could author explain how this phenomenon could affect the solutions of the model?

  • Author Response:Yes, an increase of substrate temperature would produce a rarefaction of the gases near the substrate region, thus affecting the fluxes. This effect can be easily assumed introducing the dependence of fluxes on substrate temperature. However, in PVD processes this affect would be negligible. In this work we consider PVD process.

3. Regarding the simulations, I am also concerned on the role of the substrate and its influence on the diffusion rate of deposited species. It is well known that the interaction between ad-species and the substrate can be relevant, surely for a film thickness below 10 nm. However, authors do not consider such interaction. In my opinion, this limitation should be explicitly mentioned in the text and discussed by the authors.

  • Author Response:The role of substrate is involved through the sticking coefficients between film components and substrate. That indirectly influences diffusion rate of components. Moreover, if it is necessary the diffusion coefficients of components can be assumed as concentration dependent. This reflects directly the influence of substrate material and films thickness on diffusion rate. The model has no limitation on this.

    The role of substrate is also analyzed in Figure 9 the surface roughness versus substrate temperature obtained by using three different initial conditions. Different initial condition can be interpreted as different surface pretreatment before deposition.

4. In the model, all 6 sticking coefficients are taken equal. How realistic is that?

  • Author Response:In reality sticking coefficients differ, this is important considering different interaction of different films components with substrate and between each other. We do not consider this aspect in this work and took them equal. Similar assumption was also made in our previous works and that is acceptable.

5. English should be revised. There are minor typos throughout the text (e.g. line 185 “are” is repeated twice, or in line 180 it should be “arriving at” instead of "arriving to").

  • Author Response:English is revised, mistakes corrected.

Round 2

Reviewer 2 Report

The authors have answered all my queries. I therefore suggest that the manuscript is accepted for publication.